# Field–Road Operation Classification of Agricultural Machine GNSS Trajectories Using Spatio-Temporal Neural Network

**Ying Chen** [1,2], **Guangyuan Li** [1,2], **Kun Zhou** [3,*] **and Caicong Wu** [1,2,*]

1. College of Information and Electrical Engineering, China Agricultural University, Beijing 100083, China; chenying@cau.edu.cn (Y.C.); liguangy@cau.edu.cn (G.L.)
2. Key Laboratory of Agricultural Machinery Monitoring and Big Data Application, Ministry of Agriculture and Rural Affairs, Beijing 100083, China
3. Research & Advanced Engineering, Global Harvesting, Innovation Center Randers, AGCO Corporation, Dronningborg Alle 2, 8930 Randers, Denmark
* Correspondence: kun.zhou@agcocorp.com (K.Z.); wucc@cau.edu.cn (C.W.)

**Abstract:** The classification that distinguishes whether machines are driving on roads or working in fields based on their global navigation satellite system (GNSS) trajectories is essential for effective management of cross-regional agricultural machinery services in China. In this paper, a novel field–road classification method utilizing multiple deep neural networks (MultiDNN) is proposed to enhance the accuracy of field and road point classification. The MultiDNN model incorporates a bi-directional long short-term memory network (BiLSTM), a topology adaptive graph convolution network (TAG), and a self-attention network (ATT) to effectively extract spatio-temporal features for field–road classification. The BiLSTM is used to capture temporal relationships along the time axis of a trajectory, providing global contextual information for each point. Then, the TAG network is used to obtain the spatio-temporal relationships between adjacent points in a trajectory, offering local contextual information for each point. Finally, the ATT network assigns varying weights to features to emphasize important characteristics. The performance of the MultiDNN model was evaluated using a wheat harvesting trajectory dataset, and the results showed that it achieved a high degree of accuracy, up to 89.75%, outperforming the best baseline method (GCN) by 2.79%.

**Keywords:** operation decomposition; field–road classification; deep learning; combination strategy; GNSS tracked trajectories

## 1. Introduction

The management of cross-regional agricultural machinery services in China requires accurate and up-to-date information on the operation of agricultural machinery. The service was established two decades ago to reduce farmers' mechanization costs, and has been plagued by inefficiencies in the migration of agricultural machinery, leading to overuse or underuse of the machinery [1–3]. To resolve this issue, the creation of effective migration plans based on current supply–demand information and machinery operation statistics is necessary. For example, some works [4,5] have calculated the operational costs of different in-field and out-of-field activities, so that effective migration plans can be developed to optimize cross-regional farm machinery services.

The calculation of agricultural machines' operation statistics relies on its operation classification, which is a process that breaks down the overall operation into constituent activities such as in-field effective working and non-working operation, and out-of-field activities (e.g., on-road driving). For example, Grisso et al. combined yield monitor data and GNSS trajectory data to quantify the field performance of combine harvesters and seeders, and then compared operational efficiency for each traffic mode in fields (steering, straight ahead, etc.) [6]. Bochtis et al. used a fertilizer transporter combined with its GNSS trajectory data to compare and analyze the effect of different traffic modes on operational efficiency [7].

In practice, the operation classification is often performed on GNSS recordings due to GNSS, allowing for efficient and effective data collection. Moreover, in the entire operation classification process, field–road operation classification that automatically distinguishes in-field and out-of-field activities serves as the first step. Many studies [4,5] carry out the field–road operation classification relying on an existing field boundary database; however, in China, there is no available field boundary database, making it necessary to develop a field–road operation classification method that can function without the input of field boundaries for assisting in the calculation of agricultural machines' operation statistics.

Currently, a number of methods for field–road operation classification have been presented without the need for field boundaries, which aim to automatically identify the category (field or road) of each point in GNSS trajectory. Chen et al. developed an approach for identifying field and road GNSS points based on DBCAN clustering algorithm and inference rules [8]. The DBCAN clustering algorithm is used as the first step to identify field and road GNSS points through point density (a kind of spatial relationship); then, inference rules based on the temporal relationships between points in the same field are used to correct the identification results. The DBSCAN + Rules achieved a high F1-score of 95.6% on a dataset of 60 GNSS recording trajectories collected from different tractors. Poteko et al. developed a decision-tree-based method to automatically detect field and road GNSS points [9]. The method utilizes recorded parameters such as speed and course over ground, to develop derived parameters that reflect the temporal relationship between points such as acceleration, curve radius, and angular speed. These parameters are then fed to a supervised decision tree to classify each point as either "field" or "road". Although their method achieved an accuracy over 90% on two datasets (EGNOS and the RTK dataset), it was based on high-quality GNSS data, such as the application of data correction (e.g., RTK correction, satellite-based data augmentation), and a high sampling rate (e.g., 5 Hz). Chen et al. proposed a novel method which applies graph convolutional networks (GCNs) for field–road classification [10]. This method involves constructing a spatio-temporal graph based on the relationships between each point and its neighboring points, and such relationships involve both temporal relationship and spatial relationships. The results of these studies suggest that leveraging spatio-temporal information between points is crucial for accurate field–road classification. An accuracy of 88.14% and 85.93% was achieved on two datasets, wheat and paddy, respectively. Zhang et al. proposed a novel approach for field-road segmentation that leverages multimodal information [11]. Two field–road classification results were obtained using two distinct clustering methods independently; one is DBSCAN, which partitions a trajectory based on point density, and the other is the object detection (OD) method, which clusters pixels in an image generated from the trajectory. The final result is one with a higher Davis–Bouldin Index (DBI) [12]. Their experimental results revealed that the DBSCAN + OD + DBI approach yielded superior performance compared to the individual methods, with an accuracy of 85.97%. These studies showed that leveraging spatio-temporal information between points is crucial for accurate field–road classification.

In order to improve the accuracy of "field" and "road" points classification in a GNSS trajectory, we proposed a method called MultiDNN that combines multiple deep neural networks by leveraging each's advantage to capture spatial and temporal relationships at different levels. Firstly, a bi-directional long short-term memory (BiLSTM) network, a type of recurrent neural network, was used to extract global temporal information along the time dimension by modelling a GNSS trajectory as a sequence of temporal points. BiLSTM has been shown to perform very well in time series prediction [13]. However, one-dimensional data modeling is insufficient to reveal local spatio-temporal information such as speed and the direction of distribution of points in the neighboring regions of a point. Thus, a topology adaptive graph convolution (TAG) network was used to extract local spatio-temporal information for a point [14], in which a trajectory is modeled as a spatio-temporal graph similar to the one used in Chen et al. [10]. Likewise, TAG has also been shown to achieve relatively good performance in trajectory prediction [15]. Finally, to fuse the

spatio-temporal information according to its importance, a self-attention network (ATT) was employed to emphasize the significance of the local feature [16]. BiLSTM combined with ATT has been shown to perform well in the field of natural language processing and computer vision [17,18].

The subsequent sections of this paper are structured as follows. Section 2 provides an overview of the data utilized in this study and outlines our proposed method for field–road classification. Our proposed approach utilizes a combination of BiLSTM, TAG, and ATT neural networks to effectively extract spatio-temporal features for each GNSS point. Section 3 presents a comprehensive analysis of the experimental results, which demonstrate the higher accuracy achieved by our proposed MultiDNN-based approach in comparison with the best baseline method (GCN). Finally, Section 4 summarizes our main contributions and suggests future research directions for field–road classification.

## 2. Materials and Methods

### 2.1. Dataset

Our proposed method was trained and tested using GNSS trajectory data collected by China's agricultural machinery operation big data system [19]. The dataset used in this study consists of 150 wheat harvesting trajectories, which were acquired during the wheat harvesting season (June–July) in 2021 using 65 different harvesters produced by Jiangsu Tiandian Agricultural Machinery Co. These trajectories span seven provinces in China, as shown in Figure 1.

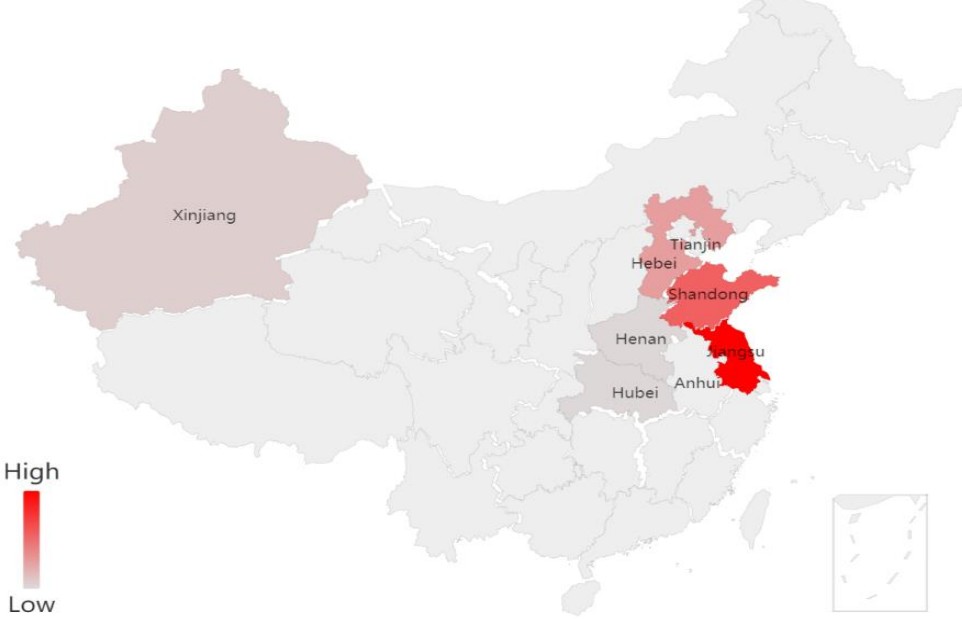

**Figure 1.** Spatial distribution of the trajectory data.

Each GNSS trajectory point is comprised of five essential parameters: longitude and latitude in the WGS84 coordinate system, velocity (in meters per second), direction (in degrees), and timestamp (in the format YYYY/MM/DD hh:mm:ss). These points were collected at an interval of approximately 5 s, and the acquisition accuracy of used GNSS receivers was 5 m (CEP). Furthermore, the collected trajectory data were manually labeled as ground truth data, representing the "field" or "road" category for each point. We observed that far fewer points were labeled as "road" than were labeled as "field", as shown in Figure 2. Specifically, the ratio of "road" points to "field" points is 1:3.97.

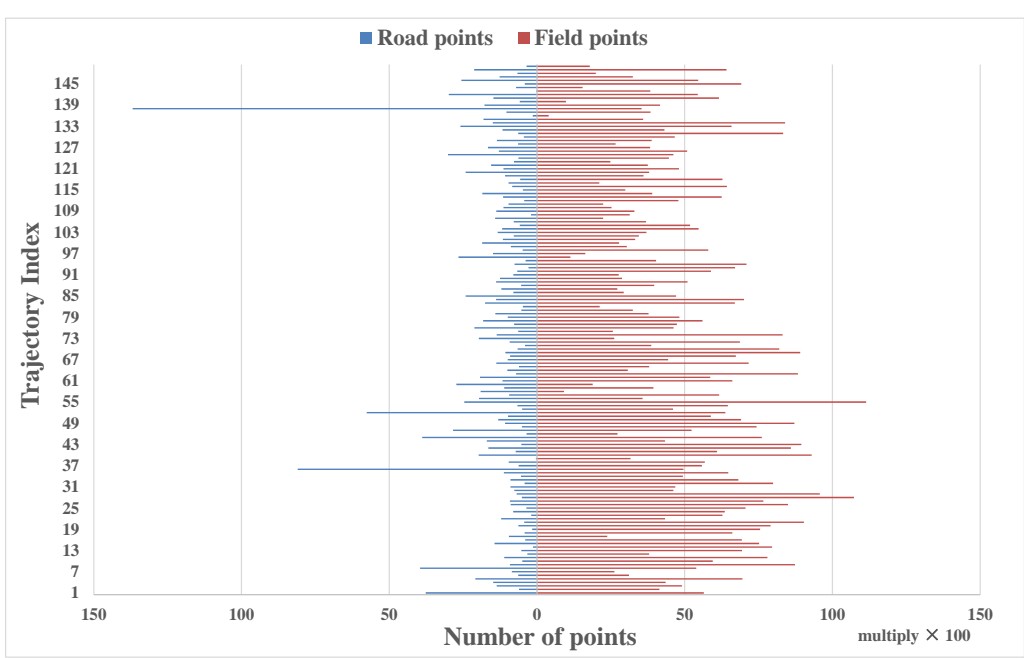

**Figure 2.** Trajectory points distribution.

## 2.2. Overview

The proposed MultiDNN method (as depicted in Figure 3) includes three components: input feature extraction; multi-layered spatio-temporal feature extraction that comprises BiLSTM for temporal feature extraction, TAG for spatio-temporal feature extraction and ATT for feature fusion; and linear classification.

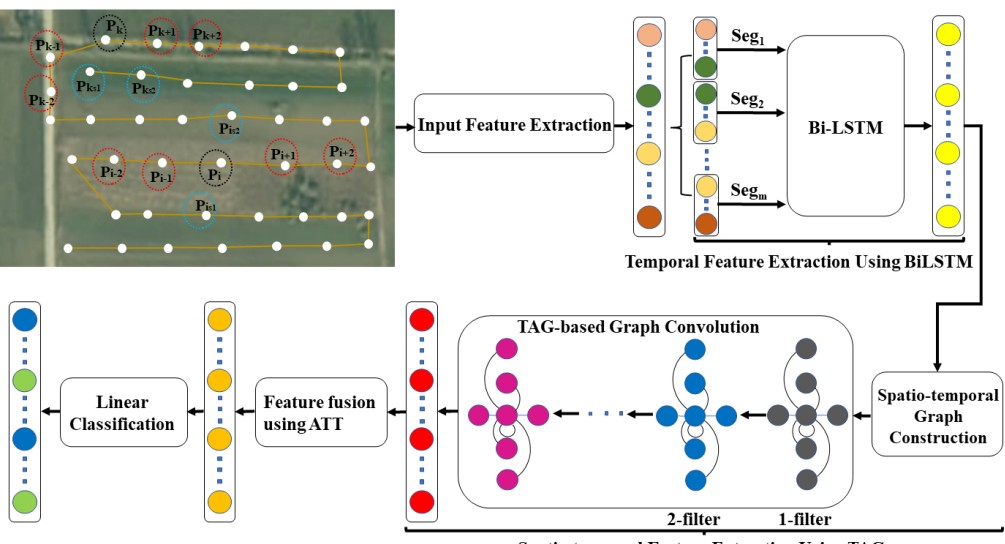

**Figure 3.** The architecture of the proposed MultiDNN-based field–road classification method, where $P_j(j = i − 2, \ldots, i, \ldots, k − 2, \ldots k + 2)$ is a GNSS point, and a trajectory is split into segments $seg_k(k = 1, \ldots, m)$ of constant size.

1.  Input feature extraction was performed to prepare the data for subsequent processing. Each point in the dataset was represented by a seven-dimensional vector, derived from the five parameters recorded for that point. This vectorization approach was adopted to enhance the subsequent feature extraction process.
2.  Temporal feature extraction using BiLSTM was performed using a BiLSTM network. This approach facilitates the extraction of global motion information for each point in

the dataset, taking into account the temporal relationships among all points within a trajectory. By modeling the sequence of points along the timeline of a trajectory, the BiLSTM network represents each point as a feature vector, which encapsulates its associated motion information.

3. Spatio-temporal feature extraction using TAG: TAG is a cutting-edge neural network that has been designed to extract spatio-temporal information by analyzing the neighboring regions of a given point. The TAG network plays a pivotal role in generating a feature vector that encapsulates the spatio-temporal relationships between the point and its neighbors. The TAG network adapts to the topology of the graph to detect spatial and temporal information, ensuring that the network effectively captures the underlying motion patterns in the data. The generated feature vector serves as the input to the subsequent ATT network.

4. Feature fusion using ATT: The ATT network employs a weighted fusion strategy based on the importance of each feature in the input vector to obtain a final feature vector. By incorporating the ATT network, certain features can be selectively emphasized or suppressed based on their relevance, thus improving the overall accuracy of the model.

5. Linear classification uses a fully connected layer and a softmax function to perform linear classification on the feature vectors generated by the ATT network. The primary objective of the classification stage was to discern the category of each point in the dataset. In other words, this classification aimed to determine whether each point belonged to the "road" or "field" category.

Moreover, input feature extraction can be regarded as a feature dimension expansion process, expanding an original five-dimensional feature vector to a seven-dimensional feature vector to represent each point in a trajectory. Next, the trajectory was segmented into constant length (e.g., 512 points), and these segments were fed into BiLSTM one-by-one to produce a new feature vector for each point based on the whole temporal information of a segment, where the segment is the one containing the point. Then, a spatio-temporal graph was constructed, which captures the spatio-temporal information of each point in a trajectory. Based on the graph and the feature vectors output from BiLSTM, a new feature vector is generated for each point using TAG-based graph convolution. Furthermore, the features in a feature vector were fused using ATT, which learns and utilizes the importance of each feature. Finally, a linear function was used to perform binary classification between the "road" and "field" categories.

*2.3. Input Feature Extraction*

To gain deeper insight into the motion, these GNSS points were transformed into a seven-dimensional vector, including speed, and six derived parameters: longitude difference, latitude difference, direction difference, acceleration, jerk, and bearing rate. The calculation method for these derived parameters can be found in the study by Dabiri and Heaslip [20].

*2.4. Temporal Feature Extraction Using BiLSTM*

BiLSTM, a popular recurrent neural network that has a strong ability to capture the temporal relationship in sequences, was utilized to extract temporal features for each point. BiLSTM, an extension of LSTM [21], utilizes two separate forward and backward hidden states to capture the past and future information for each point, and then uses the connection of the two hidden states as the temporal feature representation of that point.

Furthermore, to overcome the issue of varying lengths of trajectories in the data and the requirement of a constant length of input sequence for BiLSTM, we divided each trajectory into smaller segments of a predetermined number of consecutive GNSS points, as seen in $seg_1$ and $seg_m$ in Figure 3, where a segment is a consecutive series of GNSS points in the trajectory. If the length of the last segment was not equal to the desired length, a zero-padding was added to make all input sequences the same length. As a result of using

the BiLSTM network, each point was then represented by a feature vector that encapsulated the temporal relationships between points within the segment.

*2.5. Spatio-Temporal Feature Extraction Using TAG*

To capture the spatio-temporal relationship between a point and its neighboring points, the TAG network was utilized. As illustrated in Figure 3, there are two specific components: spatio-temporal graph construction and TAG-based graph convolution. The spatio-temporal graph construction generated a local spatio-temporal graph for each point in a trajectory, considering the spatial and temporal relationships between the point and its neighboring points. Then, taking the temporal feature representation of each point from BiLSTM as the input, the graph convolution was carried out through a topology adaptive graph convolution network (TAG), a state-of-the-art graphic neural network, which propagated the features in accordance with the graph's topology, thereby generating a feature vector for each point.

2.5.1. Spatio-Temporal Graph Construction

Based on a study by Chen et al. for a trajectory with N GNSS points [10], a graph $G = (V; E; R)$ was built based on the spatio-temporal relationship between each GNSS point and their neighbors, where $V$ represents a set of N nodes (with each node corresponding to a GNSS point), $E$ is a set of edges (with each edge signifying a relationship between two nodes), and $R$ is a set denoting the type of relationships for these edges. Specifically, there are three types of relationships: a temporal edge linking two nodes with adjacent relationships in the timeline of the trajectory; a spatial edge connecting two points with adjacent relationships in the spatial distance; and a self-loop edge connecting a node to itself. For example, as illustrated in Figure 3, node $p_i$ and its four neighbors ($p_{i-2}$, $p_{i-1}$, $p_{i+1}$, $p_{i+2}$) are connected by temporal relationship edges, and node $p_i$ and its two neighbors $p_{i_{s1}}$ and $p_{i_{s2}}$ are connected by spatial relationship edges. For further information on the construction of the spatio-temporal graph, refer to Chen et al. [10].

2.5.2. TAG-Based Graph Convolution

The topology adaptive graph convolution network (TAG) adapts to the graph topology to derive the spatial information of each node within that graph [14]. It uses a set of fixed-size filters that can be learned to perform convolutions on the graph, and these filters are designed to be adaptable to the graph's topology. Specifically, a k-size filter is used to extract the local features of a node based on its k-localized neighbor, where the k-localized neighbor is a set of nodes that can be reached from node $p_i$ through k edges in the graph. For instance, as illustrated in Figure 3, six nodes ($p_{i-2}$, $p_{i-1}$, $p_{i+1}$, $p_{i+2}$, $p_{i_{s1}}$, and $p_{i_{s2}}$) are considered 1-localized neighbors of node $p_i$ thanks to connections via both temporal and spatial relationship edges. In the end, a feature vector is generated to represent a point, which is the concatenation of all features obtained by the filters for that point.

*2.6. Feature Fusion Using ATT*

To emphasize the significance of the extracted features, the self-attention (ATT) network proposed by Vaswani et al. [16] is utilized to fuse the features by assigning weights to each feature according to its significance, resulting in a weighted representation of each point. The ATT network models the interdependencies between features, indicating the strength of association among features. Specifically, for each feature in a feature vector, a weight is learned that reflects the importance of the feature to other features. Then, a weighted feature vector is generated to represent the point.

*2.7. Linear Classification*

A fully connected layer was utilized to categorize a GNSS point, using its feature representation as the output of the ATT network. Subsequently, the softmax function proposed by Banerjee et al. was applied to calculate the predicted probability distribution

between the two categories ("field" and "road") for each point [22]. The category with the highest probability was chosen as the final predicted category for that point.

## 3. Results and Discussion

### 3.1. Experimental Settings

#### 3.1.1. Baseline Method

Our proposed methodology, MultiDNN, was evaluated against four existing field–road classification techniques, including DBSCAN + Rules [8], decision tree (DT) [9], random forest (RF) [23], and a graph convolutional network (GCN) [10]. Moreover, to show the effect of extracting temporal information along a trajectory for field–road classification, a method based only on BiLSTM was developed [13]. These state-of-the-art approaches were chosen as benchmarks to gauge the effectiveness and competitiveness of our proposed method.

#### 3.1.2. Model Training and Validation

Our evaluation methodology for the proposed field–road classification model consisted of three primary phases: data splitting, model training, and model testing. In the data splitting phase, the entire dataset was randomly partitioned into three subsets: 80% for training, 10% for validation, and 10% for testing. During the model training phase, the parameters of the selected neural network were optimized using the training data, while the performance of the trained network was evaluated on the validation set. The model that achieved the best performance on the validation set was then selected. In the final phase, the performance of the selected model was evaluated on the test data using four commonly used evaluation metrics: precision, recall, F1-score, and accuracy [24]. To ensure the reliability of our results, we repeated the experiments ten times, and reported the average performance across the ten trials.

#### 3.1.3. Implementation Details

We conducted comparative experiments between our proposed MultiDNN method and five state-of-the-art field–road classification methods: DBSCAN + Rules, DT, RF, BiLSTM and GCN. DBSCAN + Rules, proposed by Chen et al., was directly applied [8]; DT and RF were implemented by Scikit-learn (a Python-based machine learning library) [25], and GCN, BiLSTM and MultiDNN were implemented using the PyTorch framework (Python's deep learning library) [26]. Due to the differences in input requirements among these algorithms, input samples were generated according to each algorithm's specifications. Specifically, for DT and RF, an input sample is a GNSS point in the input trajectory. For DBSCAN + Rules, GCN, BiLSTM, and MultiDNN, an input sample is the entire GNSS trajectory.

In MultiDNN, the input for BiLSTM consists of segments of constant length (as described in Section 2.4), resulting in the partitioning of a trajectory into segments. Each segment (i.e., an input sample) constitutes a sequence of a predetermined number of consecutive GNSS points. In this study, the segment length and the hidden state size were set to 512 and 256, respectively. Additionally, for TAG, the number of aggregation layers in the graph convolution process was set to 8.

In the case of DBSCAN + Rules, an unsupervised learning method, the training data were used to determine the optimal input parameter settings through grid search. In contrast, for DT, RF, GCN, BiLSTM, and MultiDNN, which are supervised learning methods, the training data were used to learn the parameter values. Moreover, for the methods based on deep neural networks (DNNs), i.e., BiLSTM, GCN and MultiDNN, their model learning used the Adam optimizer with a fixed learning rate of 0.003 [27].

### 3.2. Method Comparisons and Analysis

The results of the field–road classification methods on the dataset are presented in Table 1. Among these methods, our proposed MultiDNN method achieved the best

performance, exhibiting an impressive accuracy of 89.75%. This result is notably superior to those of the baseline methods, displaying a remarkable 2.79% accuracy improvement over the best baseline method, GCN. These results suggest that our MultiDNN method is highly effective in differentiating field points from road points. The superiority of our MultiDNN can be attributed to the abilities of the three DNNs; they are capable of revealing complex spatio-temporal relationships between GNSS points, which have been proven to be important for field–road classification [13,14,16]. To be specific, BiLSTM can capture the global temporal relationship of points in a trajectory, and TAG can capture the local relationship of a point encoded in the spatio-temporal graph. Considering that the input of TAG is the results produced by BiLSTM, the TAG used in our MultiDNN can capture rich spatio-temporal information, and identifying critical features via the ATT network can further enhance the feature extraction process. The success of our MultiDNN also highlights the importance of combining different DNNs for field–road classification to exploit their strengths effectively.

**Table 1.** The overall performances of the seven methods on the data.

| Method | Pre | Rec | F1 | Acc |
|---|---|---|---|---|
| DBSCAN + Rules | 77.84 | 65.50 | 68.31 | 84.65 |
| DT | 68.16 | 52.10 | 49.48 | 81.37 |
| RF | 73.75 | 52.69 | 50.43 | 81.97 |
| BiLSTM | 81.06 | 70.11 | 72.82 | 86.34 |
| GCN | 80.66 | 72.92 | 75.14 | 86.96 |
| MultiDNN | 85.62 | 78.37 | 80.80 | 89.75 |

Pre: average precision; Rec: average recall; F1: average F1-score; Acc: accuracy.

Furthermore, the three methods based on DNNs, BiLSTM, GCN and MultiDNN, outperformed the ones based on traditional machine learning, such as DBSCAN + Rules, DT, and RF. For instance, even the weakest DNN method, BiLSTM, showed a 1.69% accuracy improvement over the best traditional machine learning method DBSCAN + Rules, indicating that the field–road classification method based on DNNs represents a promising research direction. On the other hand, among the DNN-based field–road classification methods, the accuracy achieved by BiLSTM in capturing only temporal relationships is noteworthy, reaching 86.34%. Notably, this performance is only 0.62% lower than that of GCN, which captures local spatio-temporal relationships, suggesting that temporal information along the whole trajectory is no less important than spatial information in the classification of field and road points. This indicates that the cornerstone of the high accuracy of field–road classification may rely on the method's ability to capture as many temporal and spatial relationships as possible in the trajectory data.

Moreover, Table 2 presents the comprehensive performance results of these methods on the two categories ("field" and "road") on the dataset, respectively. We observe that regardless of the method used, the classification performance of the road points was consistently lower than that of the field points. For instance, when using MultiDNN, the F1-score for the "road" category is 26.15% lower compared to the score for the "field" category. We postulate that the variation in classification efficacy between "road" points and "field" points in our study may be due to the limited availability of road points in the trajectory data (as discussed in Section 2.1). The relative scarcity of "road" points compared to "field" points may result in an insufficient number of informative samples for the model to learn from. This may impede the optimal determination of parameter settings, leading to suboptimal classification outcomes for road points. In addition, the methods based on DNNs achieved much better performances on the "road" category than the ones based on traditional machine learning. This shows that the strong model capability of these DNNs can alleviate the effect caused by the data imbalance (i.e., the limited availability of road points).

**Table 2.** The performances of the seven methods on "field" and "road" classification for the data.

| | Field | | | Road | | |
|---|---|---|---|---|---|---|
| Method | Pre | Rec | F1 | Pre | Rec | F1 |
| DBSCAN + Rules | 86.47 | 96.19 | 91.03 | 69.21 | 34.81 | 45.59 |
| DT | 82.07 | 98.79 | 89.60 | 54.25 | 5.42 | 9.36 |
| RF | 82.27 | 99.29 | 89.94 | 65.22 | 6.09 | 10.91 |
| BiLSTM | 88.15 | 96.29 | 91.97 | 73.97 | 43.93 | 53.67 |
| GCN | 89.19 | 95.67 | 92.26 | 72.13 | 50.16 | 58.03 |
| MultiDNN | 91.34 | 96.67 | 93.88 | 79.91 | 60.08 | 67.73 |

Figure 4 shows an example of the classification results of the six algorithms. It can be intuitively seen that compared to the ground truth (a), our proposed algorithm (MultiDNN) performs the best, classifying both field points and road points accurately, while the slightly lesser performing DBSCAN + Rules, BiLSTM and GCN methods have more or less misclassified trajectory points, such as identifying a small number of field points as road points (or identifying road points as field points). The worst-performing DT and RF algorithms almost completely identify road points as field points, indicating that traditional machine learning methods are effectively applicable due to the data imbalance of the dataset.

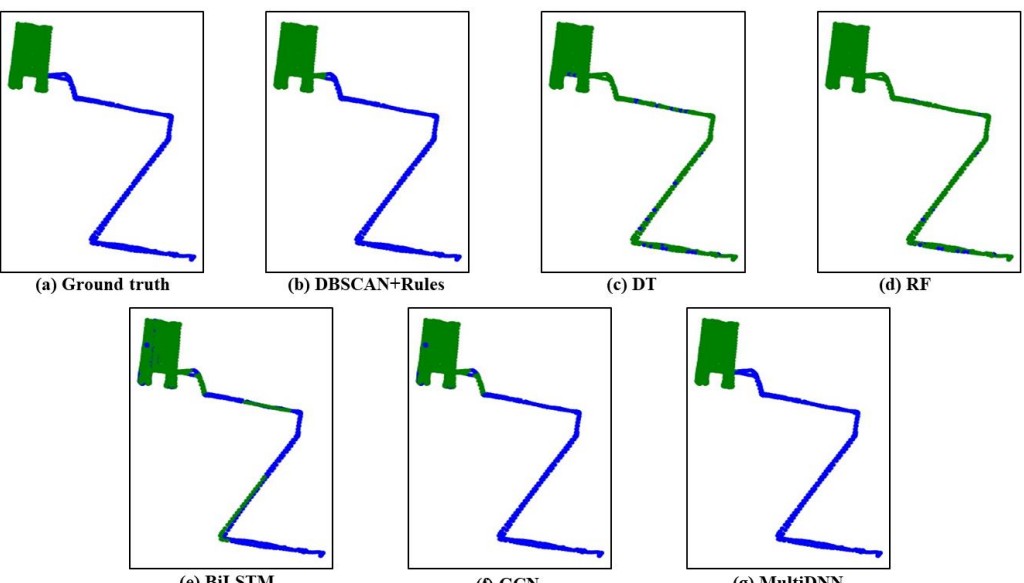

**Figure 4.** The exemplary classified results of a trajectory using selected methods.

## 4. Conclusions

This paper presents the development of MultiDNN, a novel field–road classification method that is designed to improve the accuracy of GNSS-based trajectory classification for agricultural machines. The proposed approach harnesses the power of deep learning by combining three different deep neural networks, BiLSTM, TAG, and ATT, to effectively leverage spatio-temporal information for the feature extraction of each GNSS point. Experimental results demonstrate that the MultiDNN-based field–road classification approach achieved high accuracy, with up to 89.75% for the wheat dataset, outperforming the best baseline method (GCN) by 2.79%. The findings further highlight the strong ability of deep neural networks in capturing the complex relationships between points in a GNSS trajectory. The proposed approach also showcases the effectiveness of combining different deep neural networks to improve the accuracy of field–road classification for GNSS recorded trajectories of agricultural machines.

In our future work, we aim to boost the accuracy of the MultiDNN-based field–road classification approach by exploring advanced feature representation techniques that can effectively capture both local and global information using more sophisticated deep neural networks. Additionally, we plan to investigate a robust field–road classification method that can effectively handle data imbalance in trajectories. Specifically, we intend to explore the use of data augmentation techniques and machine learning algorithms that can balance the dataset for improved classification accuracy. We believe that these efforts will further enhance the MultiDNN-based field–road classification approach, making it more effective for real-world applications in agricultural machines.

**Author Contributions:** Conceptualization, methodology, formal analysis, supervision, writing—original draft, Y.C.; methodology, data curation, investigation, validation, software, writing—original draft, G.L.; supervision, conceptualization, formal analysis, writing—review and editing, K.Z.; supervision, funding acquisition, resources, C.W. All authors have read and agreed to the published version of the manuscript.

**Funding:** This research was funded by the National Development and Reform Commission. Funding project: Integrated Data Service System Infrastructure Platform Construction Project. Grant number: JZNYYY001.

**Institutional Review Board Statement:** Not applicable.

**Informed Consent Statement:** Not applicable.

**Data Availability Statement:** As this research was funded by the project fund, the data is confidential and cannot be released for the time being.

**Conflicts of Interest:** The authors declare no conflict of interest.

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
