# Peer review of "Field–Road Operation Classification of Agricultural Machine GNSS Trajectories Using Spatio-Temporal Neural Network"

_agronomy, doi:10.3390/agronomy13051415_

Round 1
Reviewer 1 Report
In this study, a novel field-road classification method utilizing multiple deep neural networks (MultiDNN) is proposed to enhance the accuracy of field and road point classification. The findings of this study further highlight the strong ability of deep neural networks in capturing the complex relationships between points in a GNSS trajectory. Some specific problems in this paper are as follows:
1. In the literature [9]-[12], what is the accuracy of field and road identification? The main evaluation index in this paper is accuracy, and accuracy should also be used as an analysis index when studying the analysis of the current situation.
2. The introduction is too brief and does not give a detailed description of the relevant studies.
3. A more careful language revision is needed to correct the numerous mistakes that still remain.
4. In lines 81-82, the purpose of this paper is to improve the accuracy of "field" and "road" points classification in a GNSS However, the accuracy of this study does not improve compared to existing studies. Please explain the importance and necessity of this study.
5. The analysis of Tables 1 and 2 is too simplistic and does not use the data to put the key technologies into context and explain them.
6. There was no comparison with existing studies in terms of accuracy and precision. This does not explain the importance and necessity of this study.
7. Figure 4 is simply listed. No analysis has been performed on it. It also does not play any role.
8. In lines 347-355, it should not be in the conclusion, but in the limitation analysis.
Reviewer 2 Report
To distinguish whether the machines are driving on roads or working in fields based on GNSS, and improve the efficiency of cross-regional agricultural machinery, this paper proposed a novel field-road classification method combining the Bi-directional Long short term memory network, a topology adaptive graph convolution network, and a self-attention network to extract spatio-temporal features for field-road classification.
There are a few questions for author to explain:
1. As author stated in the paper that TAG is able to contract the spatio-temporal information for a point, Why is it necessary to use BiLSTM to extract the temporal information of each point in the trajectory separately?
2. The expression in Figure 3 is not clear, it is suggested to add the explanation of the block diagram.
3. How does the temporal information extracted by BiLSTM participate in the operation of TAG?
4. Line 82, it was put forward that "combines multiple deep neural networks by leveraging each's advantages", but the author did not explain each's advantages in the article or what references are cited.
5. Line 115, “the selected trajectory data were manually labeled as either ground truth data, ....... for each point”, how does the author manually mark the status of points with an order of magnitude of ten thousand?
6. In the abstract part of Line 17, the author describes TAG and ATT as two independent algorithms, but in Line126 and below, the author also puts forward the expression of TAGATT, which should be consistent in the whole paper.
